# Potential of Proteases in the Diagnosis of Bladder Cancer

**DOI:** 10.3390/cancers17213460

**Published:** 2025-10-28

**Authors:** Tomasz Guszcz, Zenon Lukaszewski, Ewa Gorodkiewicz, Adam Hermanowicz

**Affiliations:** 1Department of Urology, Hospital of the Ministry of Interior and Administration in Bialystok, Fabryczna 27, 15-471 Bialystok, Poland; 2Faculty of Chemical Technology, Poznan University of Technology, pl. Sklodowskiej-Curie 5, 60-965 Poznan, Poland; 3Bioanalysis Laboratory, Faculty of Chemistry, University of Bialystok, Ciolkowskiego 1K, 15-245 Bialystok, Poland; ewka@uwb.edu.pl; 4Department of Pediatric Surgery and Urology, Medical University of Bialystok, Waszyngtona 17, 15-274 Bialystok, Poland; ahermanowicz@wp.pl

**Keywords:** cathepsins, metalloproteinases, cancer tissue, blood serum/plasma, urine

## Abstract

Bladder carcinoma (BC) is evaluated as the ninth most common cancer worldwide and the sixth most common cancer among men. The determination of the occurrence and stage of the disease is a significant diagnostic task. An alternative to an invasive biopsy may be the determination of biomarkers in patient samples such as bladder tissue, blood serum or plasma or urine samples. The aim of this paper is to review reports on the role of proteases in bladder cancer and their determination in cancerous samples. It is concluded that the reviewed papers do not provide a clear picture concerning the use of proteases as bladder cancer biomarkers, or concerning the levels of particular proteases in control samples. More work in this area is necessary, especially by scientists equipped with new analytical tools.

## 1. Introduction

Bladder carcinoma (BC) is evaluated as the ninth most common cancer worldwide and the sixth most common cancer among men [1]. Non-muscle invasive BC (including stages Ta and T1), represents 80% of cases, while muscle invasive BC (stages T2, T3, and T4), accounts for the remaining 20% [2].

The urinary bladder collects urine produced by the kidneys. The bladder itself consists of three layers: the muscular layer, the submucosa, and the mucosa. The bladder is covered on the inside by the urothelium and on the outside by the serosa. These layers are used to indicate how deeply the cancer has invaded and to identify the stage of the disease. Absence of bladder cancer is denoted as T0. An invasion of cancer onto the surface of the urothelium is classified as bladder cancer stage 1 (Ta or Tis) [3,4]. Penetration of cancer into the urothelium is stage 2 (T1). Both Ta and T1 are classified as non-muscle invasive bladder cancer. The subsequent stages of bladder cancer reflect stages of the penetration into the muscle (T2, T3) or through the bladder muscles (T4) [4].

Determination of the occurrence and stage of bladder cancer is a significant diagnostic task. The determination of biomarkers in patient samples may be an alternative to an invasive biopsy. Apart from bladder tissue, samples may be of blood serum or plasma, or urine. Blood and urine samples are highly available. Urine has been in close contact with growing bladder cancer and seems to contain very valuable information, including biomarker concentration [5,6,7,8]. However, the problem is the availability of sufficiently sensitive analytical methods. The detection of bladder cancer is only the first task expected to be performed based on biomarker concentrations in body fluids; the next is the stage determination of the cancer. The availability of analytical methods that enable the detection of bladder cancer in body fluids will provide an opportunity for the wide screening of the population for this disease.

The aim of this paper is to review reports on the role of proteases in bladder cancer and their determination in cancerous samples. Bastos et al. [5] show that some proteases may be prospective biomarkers of BC. Proteases are a family of enzymes that have the ability to cleave peptide bonds in proteins into smaller peptides or single amino acids [9]. They are present in lysosomes, where their function is to degrade proteins, but not to destroy cells, rather to maintain cellular homeostasis and recycle cell constituents [10]. Apart from an intracellular-degrading role, they participate in various biological processes as key signalling molecules [10]. Proteases catalyse irreversible hydrolytic reactions; therefore, their proteolytic activity must be strictly regulated by pH, temperature, oxidation, and protein inhibitors.

Proteases can be classified in several groups depending on their catalytic residue: aspartic, cysteine, serine, metalloproteinases, as well as threonine and glutamic [11]. Aspartic proteases include cathepsins D and E. The most numerous group is the cysteine proteinases, to which belong, among others, cathepsins B, C, F, H, K, L, O, S, V, X, and W [12]. These are the 11 human cysteine cathepsins, existing at the sequence level; this was confirmed by a bioinformatic analysis of the draft sequence of the human genome. Matrix metalloproteinases (MMPs) belong to a broader group of zinc endopeptidases called metzincins [13]. MMPs include gelatinases (MMP 2 and MMP 9), stromelysins (MMP 3, MMP 10 and MMP 11), membrane-type MMP (among others MMP 14, MMP 15), matrilysin (MMP 7, MMP 26), and collagenases (among others MMP1). Their significant function is the degradation of components of the extracellular matrix. An emerging group of proteases involved in bladder cancer progression is the ubiquitin-specific proteases. The existing literature concerning proteases available on Scopus is analysed below. A majority of reviewed papers contains patient and tumour characteristics.

## 2. Cathepsins

Cathepsin D (Cath D) is a lysosomal aspartyl protease, having an MW of 48 kDa, consisting of two chains of 34 kDa and 14 kDa [14]. Cath D has isoforms and is formed from procathepsin D (52 kDa). The isoforms have isoelectric points (pI) between 5.5 and 6.6 and optimum pH between 2.8 and 5.5. Several papers have reported higher Cath D concentration in bladder cancer tissue than in normal bladder tissue [15,16,17,18,19]. According to Ozer et al. [20], Cath D in tumour tissue has no predictive value. Gorodkiewicz et al. [21] reported that Cath D concentration in the blood serum and urine of patients suffering from bladder cancer is higher than in healthy subjects (approximately 3 ng/mL for cancer and 0.5 ng/mL for controls, in both for serum and urine). Guszcz et al. [22] reported that Cath D activity in the blood serum of patients suffering from bladder cancer is higher than in the control group. The quotient of urine Cath D and total protein concentration is better suited as a bladder cancer biomarker [21]. Most of these results were obtained using immunohistochemical methods [18,19]; however, the methods of immunoradiometric assay [15] and array SPRi [21] were also used.

Cathepsin B (CathB) is a lysosomal cysteine proteinase synthesised as a zymogen of 39–47 kDa, which is subsequently converted into an active single-chain form of 33 kDa (CB33) and, by additional processing, into an active 2-chain form containing a heavy chain of 27–29 kDa (CB27-29) and a light chain of 4–6 kDa [23]. A high concentration of Cath B in transitional bladder cancer tissue and in plasma membranes correlates with the invasion of the disease [22,23,24,25]. An increase in Cath B concentration has been shown to enhance the invasive ability of cancer cells in in vitro experiments [26]. As regards body fluid concentrations, the high diagnostic potential of Cath B in urine has been reported [23]; however, there are also reports of a lack of correlation between Cath B concentration and the disease outcome variables [25,27]. The median Cath B concentration in the urine of transitional bladder cancer patients was reported as 5.9 µg/mL against 3.8 µg/mL for a control group [23], while another study gave 3.87 µg/mL for bladder cancer patients against 1.35 µg/mL for the control group [28]. According to Kotaska et al. [28], the Cath B/creatinine ratio is superior for diagnostic purposes to simple Cath B concentration. Apart from Cath B, urine precathepsin B is also a valuable biomarker of bladder cancer [29]. The diagnostic value of serum Cath B concentration in bladder cancer diagnosis is low [29]. Cath B concentration in urine and serum was determined using commercially available enzymatic immunoassays [28,29] or a spectrofluorimetric assay [25], while Western blot analysis was also used in the case of tissue [23].

Cathepsin L (Cath L) is a lysosomal cysteine-type endopeptidase, acting similarly to papain and Cath B, having higher specificity towards proteins than other cathepsins [14]. It is a dimer composed of heavy and light chains linked by disulfide bonds, and has an MW around 35–50 kDa [30]. Cath L concentration in the urine of patients with superficial and muscle-invasive bladder tumours has been shown to be much higher than in a control group [25]. Using standard immunochemistry, Yan et al. [31] localised Cath L in the cytoplasm of malignant cells of bladder cancer and found that it is strongly associated with the invasiveness of the tumour. However, another study [32] did not find statistically significant differences between cancerous bladder tissue and normal tissue. Standard immunohistochemistry was used for the determination of Cath L in tissue [31,32], and a spectrofluorimetric assay for Cath L determination in urine [25]. Recently, an array SPRi method for the determination of Cath L in cancerous samples was published [33]. This method is developed for Cath L determination in body fluids with regard to specificity, signal linearity range and small amount of sample.

Cathepsin H (CatH) is a lysosomal cysteine protease with a unique aminopeptidase activity. It consists of a single polypeptide chain of 28 kDa. The enzyme exists in multiple isoelectric forms, the two major forms having pI values of 6.0 and 6.4 [14]. A neutral pH level is optimal for CatH activity, and it is therefore expected to be active in the extra-lysosomal and extracellular space. In cancerous bladder tissue, Cath H concentration was reported to be significantly higher (*p* < 0.05) than in matched normal tissue [32].

Cathepsin V, also known as cathepsin L2, is a type of lysosomal cysteine protease that belongs to the papain family [34]. Its MW is 37 kDa, and it exhibits maximum activity at pH 5.7, being unstable at neutral pH. CathV is reported to participate in the development and progression of bladder cancer and may be used as a biomarker and potential target in the study of the disease [35]. Bladder cancer patients with high CathV levels had considerably poorer overall and disease-free survival rates than patients with lower levels of CathV, suggesting a potential role of CathV in the development and progression of bladder cancer. Overall, high CathV expression was associated with a poor prognosis in bladder cancer patients [35]. The expression of Cath V in lysates of bladder cells was determined by the Western blot method [35].

Cath S has an indirect influence on bladder cancer tumours, because it reduces the immune suppressive activity of T-cells [30]. This enzyme remains catalytically active at neutral pH and has a pH optimum range between 6.0 and 7.5.

Cath C, also known as dipeptide base peptidase I, is an exopeptidase having an M.W. of 210 kDa. It is maximally active at acidic pH, requiring a thiol cofactor [14]. Its concentration is elevated in bladder cancer tissues and cells, and is associated with poor survival prognosis. Cath C enhances the activity, proliferation, migration, and invasion of bladder cancer cells via an increased DIAPH3 and activating Wnt/β-catenin pathway [36].

## 3. Metalloproteins

Matrix metalloproteinase 2 (MMP-2), also known as gelatinase-A or neutrophil gelatinase, is a protease of 66 kDa. It degrades gelatin, fibronectin, laminin, collagens I, II, III, IV, V, VII, X, and XI, and also aggrecan, elastin, and proteoglycans [37]. MMP 2 is involved in angiogenesis, vascular remodelling, tissue repair, cancer invasion, and inflammation [13].

The ability of MMP2 to degrade collagens is significant in the case of bladder cancer, because collagens are among the main components of the urinary bladder [38]. Based on the analysis of several databases, Shen et al. [39] concluded that MMP2 and several other MMPs are possible prognostic biomarkers for patients with bladder cancer. The expression of MMP2 changes during the development of bladder cancer [38,40,41]. The activity of MMP2 was determined as 8897 ± 1655 1/(mg/mL) in bladder transitional cell cancer of grade I, 18,355 ± 5307 1/(mg/mL) in BTCC of grade II, and 26,467 ± 4705 1/(mg/mL) in BTCC of grade III [42]. Urinary concentration of MMP2 is a promising biomarker of bladder cancer [5,43] and allows discrimination between BC patients and healthy subjects; however, it failed in accurate BC identification in the early stages of the disease. Plasma concentration of MMP 2 in bladder cancer patients is between 406 and 3751 µg/L, while in healthy subjects it lies between 547 and 1295 µg/L [44]. The participation of MMP2 in various pathways related to bladder cancer has been reported in a range of papers [45,46,47,48,49]. MMP2 was determined in blood plasma or in cancer cell culture using the ELISA kit [44,49,50] and in bladder cancer tissue by means of gelatine zymography [40,44]. It can also be determined by the array SPRi [51].

Matrix metalloproteinase 9 (MMP-9), also known as gelatinase-B, is a zinc-dependent metalloproteinase with an MW of 86 kDa in its active form. It participates in the remodelling of the extracellular matrix in various pathological states such as angiogenesis, vascular remodelling, tissue repair, cancer invasion, and inflammation [13] by degrading gelatine, collagens IV, V, VII, X, and XIV, fibronectin, laminin, aggrecan, and elastin [36]. Because collagens are among the main components of the urinary bladder [37], the ability of MMP9 (along with other MMPs) to degrade collagens is a significant factor in urinary bladder cancer. Based on an analysis of several databases, Shen et al. [38] concluded that MMP9 and several other MMPs are possible prognostic biomarkers for patients with bladder cancer. There have been several reports that the expression of MMP 9 changes during the bladder cancer development [37,39,40,52] and that MMP9 and its dimer are overexpressed in high-grade bladder cancer samples in comparison with samples taken from patients with a lower grade [5,53,54]. The serum MMP 9 concentration of bladder cancer patients ranges from 4.4 to 412 (av. 22.9) µg/L, but in control groups it lies in a very similar range of 3.5–411 (av. 19.4 µg/L). According to Bastos et al. [5], MMP 9 from the urine of patients suffering from bladder cancer is a prospective biomarker of the disease. The biomarker is determined by immunostaining (e.g., [54]), or by the sandwich ELISA technique, e.g., [44]. Comparing MMP2 and MMP9, as potential bladder cancer biomarkers, both are suitable for T2 or higher stages of the disease, while both have failed at early stages [37]. Both of these metalloproteases belong to the same group of gelatinases.

Several other MMPs are also considered as bladder cancer biomarkers: MMP1, MMP3, MMP7, MMP14, and MMP15. MMP14 and MMP15, both belonging to the group of membrane-type proteinases, were compared as bladder cancer biomarkers [55]. Matrix metallopeptidase 15 (MMP15, also MT2-MMP, MT2MMP, MTMMM2 or SMCP-2) has MW 72 kDa [56]. Like other MMPs in this group, it degrades collagen types I, II, and III, aggrecan, elastin, fibronectin, gelatine, laminin, MMP-2, and MMP-13. MMP15 is highly expressed in bladder cancer tissue and contributes to inflammation and angiogenesis in bladder cancer [57]. It may promote cancer cell invasion into the basement matrix through MMP 2 activation. The total content of MMP15 in the tissue of low-grade bladder cancer is 36.638 ± 4.895 µg/mg protein, while in the tissue of high-grade bladder cancer it is lower, at 22.974 ± 3.042 µg/mg protein. In control bladder tissue, the total content of MMP15 was measured at 27.793 ± 3.985 µ/mg protein [55]. The concentration of MMP 15 was evaluated using the enzyme-linked immunosorbent assay (ELISA) [55].

Metallopeptidase 14 (MMP 14), also known as membrane-type 1 matrix metalloproteinase (MT1-MMP), is a molecule of 60 kDa in its active form [58]. It degrades collagens type I, II, and III, promoting angiogenesis, cellular invasion in cancerogenic processes, and metastasis [59,60]. An elevated concentration of MMP14 appears to be associated with a high degree of malignancy, aggressiveness, and poor survival prognosis [56,61]. The total content of MMP14 in bladder tissue depends on the stage of bladder cancer: it is reported at 10.133 ± 1.414 µg/mg protein for a low grade of the disease and 81.784 ± 9.876 µg/mg protein for a high grade, compared with 7.454 ± 1.183 µg/mg protein for control samples of bladder tissue [55]. The enzyme-linked immunosorbent assay (ELISA) was used for the determination of MMP 14 [55].

Kudelski et al. [55] compared the use of MMP14 and MMP15 for the determination of the stage of bladder cancer based on their concentration in the cancer tissue. They concluded that MMP14 is superior to MMP15. The level of MMP14 grows significantly with the grade of bladder cancer, while changes in MMP15 content in bladder cancer tissue, depending on the grade of the disease, are inconsistent.

Matrix metalloproteinase 10 (MMP 10), also known as stromelysin-2, SL-2, STMY2 or Transin-2, is a molecule consisting of 476 amino acids, having an MW of 54 kDa. It degrades collagen types III, IV, and V as well as aggrecan, elastin, fibronectin, gelatine, laminin, MMP-1, and -8 [50]. The role of MMP 10 in bladder cancer is ambiguous. According to Seargent et al. [62] the expression of MMP-10 likely does not show any relationship with the progression of urinary bladder cancer. However, Kudelski et al. [50] reported that the content of MMP-10 was significantly lower in control tissue (2.75 µg/g protein) than in low-grade (4 µg/g protein) and high-grade (4.25 µg/g protein) cancer tissues (*p* < 0.001 in both cases). MMP 10 was determined using the ELISA and Western blot techniques.

Matrix metalloproteinase 7 (MMP7), also known as matrilysin-1, is a zinc- and calcium-dependent endopeptidase of 30 kDa, degrading components of the extracellular matrix [63,64]. MMP7 participates in the signalling of the miR-133a-3p/SLC12A5/SOX18/MMP7 axis, which promotes the progression of urothelial carcinoma [65].

Matrix metalloproteinase 3 (MMP3), also known as stromelysin-1, SL-1, STMY 1, or Transin-1, has an MW of 54 kDa. It degrades gelatine, elastin, aggrecan, proteoglycans, laminin, fibronectin, collagen types II, III, IV, IX, X, XI, as well as metalloproteins MMP-7, 8, and 13 [66,67,68]. MMP 3 also exhibits the ability to activate inactive metalloproteinases. The content of MMP3 in bladder cancer tissue (0.2 µg/g protein) is significantly lower than in control bladder tissue (1.8 µg/g protein) [50]. MMP 3 concentration in the plasma of metastasised bladder cancer patients is significantly higher (17.7 ng/mL) than in plasma from healthy donors (9.9 ng/mL) [44]. The concentration of MMP3 was measured using the ELISA [50].

Matrix metalloproteinase 1 (MMP1), also known as interstitial collagenase or fibroblast collagenase, is an enzyme of ca. 54 kDa. It belongs to the group of collagenases. It cleaves collagen types I, II, III, VII, and X, enabling cancer progression [68]. Serum concentration of MMP 1 decreases with bladder cancer progression, and has been reported at 2.8 ng/mL in bladder cancer patients compared with 5.6 ng/mL in a control group [44]. The concentration of MMP 1 was measured by means of the ELISA, but it can also be measured by array surface plasmon resonance [69].

## 4. Ubiquitin-Specific Proteases

Ubiquitin-specific proteases (USP) are deubiquitination enzymes which remove ubiquitin from ubiquitinated proteins, reversing ubiquitination. Several of them participate in bladder cancer development. Ubiquitin-specific protease 1 (USP1) has been found to be upregulated in human bladder cancer cells and to correlate with poor patient prognosis [70]. USP 1 deficiency inhibited tumour formation in in vivo experiments with human bladder cancer cells UMUC3. Ubiquitin-specific protease 2a (USP2a) has oncogenic properties. It caused enhanced proliferation in immortalised TRT-HU1 normal human bladder epithelial cells [71]. USP5 protein levels are significantly elevated and positively associated with Twist1 levels in clinical bladder cancer samples [72]. Higher expression of USP5 protein was observed in a T24 cell line (representing a human bladder cancer cell line) than in a non-tumorigenic urothelial cell line SV-UAC 1 [73].

## 5. Discussion

A summary of the analysed studies on the relationship between the occurrence of bladder cancer and the concentration of particular markers is provided in Table 1. Additional information recorded includes the matrix and the method of biomarker determination.

The majority of the reviewed papers report an increase in the marker concentration in bladder cancer samples versus the control. Only in the cases of MMP1 and MMP10 do the bladder cancer samples have lower marker concentrations than the control samples. In the case of the biomarkers MMP3 and MMP15, the results are inconclusive. In the case of Cath B, two papers [23,28] reported an increase in Cath B concentration in samples from bladder cancer patients, while according to one paper [29], the results are inconclusive.

Above half of the reviewed papers investigated the concentration of markers in bladder cancer tissue. However, only a few of them provided quantitative data. These values are given in Table 2.

Only a few papers provide information about MMP biomarker concentrations in bladder tissue (cancerous and control tissue), and none provide such information about cathepsins. It is notable that most of these data appear only in relatively recent papers, which can be assumed to be related to progress in analytical techniques. Levels of µg/g protein are characteristic of most MMP biomarkers in urinary bladder tissue. In the case of MMP 2, different units were applied. Anyhow, progress in MMP 2 concentration following the cancer progression is clear.

More of the analysed studies investigated the marker’s concentrations in body liquids: blood serum and urine. Reported results for comparisons of biomarker levels in the body fluids of bladder cancer patients and control samples are listed in Table 3.

Most of the reported concentrations of proteases in blood serum/plasma and urine are of the order of ng/mL, in both cancerous and control samples. Only MMP2 concentration in plasma is higher than the other proteases [44]. Generally, protease concentrations in cancerous samples are higher than in the controls. The only exception was MMP1 in plasma, where the control results were higher than those for samples from cancerous patients.

The number of reviewed papers providing quantitative information about protease concentrations in bladder cancer and control samples is relatively low. In total, six cathepsins and eight MMPs are reported, but quantitative data are provided for only six proteases in body fluids and four proteases in bladder tissue. Moreover, almost all data are represented by a single source, whereas confirmation from a minimum of two sources is highly desirable. Biomarker concentrations in serum/plasma increase in the order: Cath D < MMP 1 < MMP 3 < Cath B < MMP 2.

Several studies attempted to link biosensor concentration, mostly in cancerous tissue, with the stage of bladder cancer. The greatest difference related to the stage of the disease was observed by Kudelski et al. [55] in the case of MMP 14, where the concentration in tissue from a low grade of bladder cancer was approximately 10 µg/mg, while for a high stage of the cancer it was approximately 82 µg/mg. However, in the case of MMP 15, the results were inconclusive, and similarly for MMP 10 [50] because the differences in the results were small. In the case of MMP 2 there a gradual growth in the biomarker concentration in cancerous depending on the cancer stage was observed; it was 8897 1/(mg/mL) in stage I of the cancer, 18,355 1/(mg/mL) in stage II, and 26,457 1/mg/mL in stage III [42].

The blood plasma concentration of MMP 9 was reported to increase gradually depending on the stage of the bladder cancer [44]: plasma from patients having metastasised bladder cancer contained approximately 56 ng/mL, while plasma from patients having a lower stage of the disease contained approximately 23 ng/mL; control sample concentration was approximately 19 ng/mL. The results for plasma MMP 1, MMP 2, and MMP3 were inconclusive.

Apart from blood plasma/serum and urine, exosome may provide valuable information concerning bladder cancer [74], as well as gene methylation [75]. A new trend in the investigation of the participation of proteases in bladder cancer involves in vitro experiments with immortalised bladder cancer cell lines such as T24 and normal bladder cell lines such as TRT-HU1.

It should be noted that almost all of the quantitative results were obtained using the ELISA technique [28,44,50,55], while the results obtained with the immunohistochemical method, the immunoradiochemical method, gelatine zymography or Western blot are only semiquantitative. Commercially available ELISA kits for MMP 9 determination have different ranges of measurable concentration, 0.3–20 ng/mL to 1–30 ng/mL, while a homemade kit offers a range of 0.015–2.0 ng/mL [76]. The precision of measurements is 2.5% or 3.7%, depending on the kit manufacturer. Taking into account concentration ranges of particular biomarkers, ELISA is suitable for the determination of these biomarkers in body fluids, after appropriate dilution in some cases.

Very credible data were also obtained in research using the array SPRi technique [21]. This technique enables the determination of MMP1 [73], MMP2 [51], Cath D [24], Cath G [77], Cath L [33], and Cath S [78], with limits of quantification below the order of ng/mL, and thus suitable for the determination of these biomarkers in body fluids after appropriate dilution, and precision of order 5% depending on a particular biosensor.

Extremely low detection limits of MMP of order of fM and aM can be achieved using an electrochemiluminescence biosensor integrated with T7 RNA polymerase amplification and CRISPR/Cas13a-mediated signal enhancement [79,80]. A detection limit of order fg/mL can be achieved by an electrogravimetric biosensor [81]. However, such low detection limits are hardly necessary for the determination of MMP in body fluids, considering the more complicated measuring procedure. The LC-MS technique may also be suitable for MMP determination [82].

It is concluded that the reviewed papers do not provide a clear picture concerning the use of proteases as bladder cancer biomarkers or concerning the levels of particular proteases in control samples. More work in this area is necessary, especially by scientists equipped with new analytical tools mentioned above. The development of new biosensors for the determination of MMP and USP is also desirable. The accumulation of a sufficient number of credible results may be a basis for the development of an algorithm for the determination of bladder cancer stage and subsequently its introduction into diagnostics.

## Figures and Tables

**Table 1 cancers-17-03460-t001:** Effect of bladder cancer on the concentrations of particular biomarkers.

Biomarker	Medium	Tumour vs. Control	Technique	Reference
Cath D	tissue	+	immunohistochemical	[19]
Cath D	tissue	+	immunoradiometric assay	[15]
Cath D	serum; urine	+	Array SPRi	[21]
Cath B	tissue	+	Western blot ELISA	[22,23,31]
Cath B	urine	+	Western blot	[22]
Cath B	urine	±		[27]
Cath B	serum	+	ELISA	[28]
Cath L	urine	+	spectrofluorimetric	[24]
Cath L	tissue	d	immunohistochemical	[31]
Cath L	Tumour cells cytoplasm	+	immunohistochemical	[30]
Cath L	urine	+	spectrofluorimetric	[24]
Cath H	tissue	+	immunohistochemical	[31]
Cath V	tissue	+	Western blot	[34]
MMP2	tissue	+	gelatine zymography	[40]
MMP2	plasma	+	ELISA	[44]
MMP9	tissue	+	immunohistochemical	[5,52,53]
MMP15	tissue	±	ELISA	[55]
MMP14	tissue	+	ELISA	[55]
MMP10	tissue	d	Western blot ELISA	[50]
MMP3	tissue	±	Western blot ELISA	[50]
MMP3	plasma	+	ELISA	[44]

+ denotes an increase in the marker concentration in case of bladder cancer; d denotes a decrease, and ± denotes that the paper does not provide a clear answer.

**Table 2 cancers-17-03460-t002:** Biomarker concentrations in bladder tissue.

Biomarker	Control	Bladder Cancer: Stages			Reference
		I	II	III	
MMP2	n.d.	88,896 ± 1655 *	18,355 ± 5307 *	26,465 ± 4705 *	[42]
MMP3	1.75 ± 0.20 **	0.25 ± 0.20 **		1.2 ± 0.10 **	[50]
MMP10	2.75 ± 0.25 **	4.0 ± 0.50 **		4.2 ± 0.50 **	[50]
MMP14	7.50 ± 1.18 **	10.1 ± 1.41 **		81.8 ± 9.88	[55]
MMP15	27.8 ± 3.98 **	36.6 ± 4.90 **		23.0 ± 2.94 **	[55]

* (mg/mL)^−1^ = mL/mg; ** µg/g protein.

**Table 3 cancers-17-03460-t003:** Biomarker’s concentration in the body liquids of bladder cancer patients and controls.

Biomarker	Medium	Control [ng/mL]	Bladder Cancer [ng/mL]	Reference
CathD	serum	0.28–0.52	1.3–5.59	[21]
MMP9	plasma	19.4	56.3	[44]
MMP3	plasma	11.9	17.7	[44]
MMP2	plasma	550–1300	410–3750	[44]
MMP1	plasma	5.6	2.8	[44]
CathB	serum	4.3–102		[28]
CathB	urine	1.35	3.87	[28]
CathD	urine	0.32–0.68	1.35–7.14	[21]

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
