# Peer review of "Potential of Proteases in the Diagnosis of Bladder Cancer"

_cancers, 2025, doi:10.3390/cancers17213460_

Round 1

Reviewer 1 Report (Previous Reviewer 3)

Comments and Suggestions for Authors

We appreciate the review of the manuscript and the addressing of most of the comments raised in the first round of review.

The introduction has been substantially improved, particularly with the updated incidence and prevalence data, and the discussion is clearer and more comprehensive. The inclusion of additional proteases partly justifies maintaining the original title.

However, some important issues remain:

  1. Table 2: This table continues to present inconsistencies. Only a few MMPs are included, the values are approximate rather than precise, and only low-grade data are included. In contrast, the main text presents inaccurate values and distinguishes between low- and high-grade tumors. These discrepancies should be corrected to ensure consistency and accuracy. The authors should revise the table so that it reflects the data described in the corresponding section, with the appropriate statistical descriptors (mean/median ± SD or SE).
  2. Quantitative Data in the Discussion: In several cases, values are reported as "approximately" even though they are expressed to three decimal places (e.g., approximately 9,000 mg/mL). This is misleading. If values are approximate, they should be rounded accordingly; if they are precise, the term "approximately" should be removed.

Overall, the manuscript has improved, but special attention to numerical precision and consistency between the text and tables is still needed. Addressing these issues will significantly strengthen the scientific rigor of the manuscript.

Author Response

However, some important issues remain:

    Table 2: This table continues to present inconsistencies. Only a few MMPs are included, the values are approximate rather than precise, and only low-grade data are included. In contrast, the main text presents inaccurate values and distinguishes between low- and high-grade tumors. These discrepancies should be corrected to ensure consistency and accuracy. The authors should revise the table so that it reflects the data described in the corresponding section, with the appropriate statistical descriptors (mean/median ± SD or SE).

    Quantitative Data in the Discussion: In several cases, values are reported as "approximately" even though they are expressed to three decimal places (e.g., approximately 9,000 mg/mL). This is misleading. If values are approximate, they should be rounded accordingly; if they are precise, the term "approximately" should be removed.

Reviewer 2 Report (Previous Reviewer 2)

Comments and Suggestions for Authors

This review aims to consolidate existing findings on the role and quantification of proteases in bladder cancer (BC), with a focus on their potential as non-invasive biomarkers. The topic is highly relevant and scientifically valuable, especially in the context of early detection and monitoring of BC, the manuscript requires revision to improve clarity, depth of analysis, and overall impact.

  1. The introduction lacks a structured flow. The clinical significance of bladder cancer should be better contextualized with recent incidence and mortality data (WHO, GLOBOCAN, etc.).
  2. Clearly define the rationale for focusing on proteases. Why are they prioritized over other classes of biomarkers like miRNAs or EVs?
  3. The manuscript currently provides a descriptive rather than critical or comparative assessment of the literature. Consider adding:
  4. A table summarizing studies: type of protease, sample type, method of detection, fold change vs. control, and sample size.
  5. Discussion on inconsistencies across studies, e.g., differences in assay sensitivity, sample preparation, or patient demographics.
  6. Incorporate discussion of emerging technologies for protease detection (e.g., mass spectrometry, biosensors, immunoassays with nanomaterials).
  7. The key finding that most studies lack quantitative data on cathepsins is important. However, this insight should be supported by a systematic breakdown: how many studies included quantification? Which methods were used? What detection limits were reported?
  8. Clarify units (µg/g tissue protein vs. ng/mL in fluids) and provide context—e.g., what threshold levels might be clinically relevant?
  9. Some important recent studies on MMPs and cathepsins in cancer diagnostics seem to be missing. Ensure inclusion of recent (2022–2024) high-quality publications, especially from clinical or translational research journals.
  10. Mention potential for clinical translation or integration into multi-marker diagnostic panels.
  11. The following studies are suggested to evaluate and add in the literature review of manuscript: https://doi.org/10.1016/j.jpba.2023.115937, https://doi.org/10.3389/fonc.2024.1432869,10.1080/17435889.2025.2493037

The author has successfully addressed all the comments and suggestions provided in the previous review. The revisions and editions made are satisfactory and have enhanced the overall quality of the manuscript.

Author Response

The author has successfully addressed all the comments and suggestions provided in the previous review. The revisions and editions made are satisfactory and have enhanced the overall quality of the manuscript.

We express our thanks for the mansuscript approval.

Reviewer 3 Report (Previous Reviewer 1)

Comments and Suggestions for Authors

1- The abstract merely summarizes existing literature without any original data, insights, or analysis.

2- There is no evidence of a systematic review protocol or meta-analytic approach to synthesize the reviewed studies.

3- The scope of reviewed studies is unclear; no mention of search databases, inclusion/exclusion criteria, or study timeline.

4-Despite reviewing numerous proteases, the authors fail to propose specific biomarkers with clinical utility.

5- The review does not explore or discuss conflicting findings.

6- Quantitative information is notably missing for most proteases.

7- The biological significance of the elevated protease levels is not discussed in the context of bladder cancer progression or pathogenesis.

8-There is no discussion of standardization issues across studies (e.g., sample type, assay technique, or patient heterogeneity).

9- The authors fail to correlate protease levels with bladder cancer stages, grades, or prognosis.

10- Key challenges in translating protease biomarkers into clinical diagnostics (e.g., reproducibility, validation, regulatory approval) are ignored.

Not all comments have been properly addressed. The revised file lacks figures and does not include a graphical abstract. The revisions appear superficial, and I therefore recommend rejection.

Author Response

Not all comments have been properly addressed. The revised file lacks figures and does not include a graphical abstract. The revisions appear superficial, and I therefore recommend rejection.

We hope that our paper after secondary revision is acceptable. We hope that figures and graphical abstract are not obligatory in Cancers.

Round 2

Reviewer 3 Report (Previous Reviewer 1)

Comments and Suggestions for Authors

Accepted for publication 

This manuscript is a resubmission of an earlier submission. The following is a list of the peer review reports and author responses from that submission.

Round 1

Reviewer 1 Report

Comments and Suggestions for Authors

1- The abstract merely summarizes existing literature without any original data, insights, or analysis.

2- There is no evidence of a systematic review protocol or meta-analytic approach to synthesize the reviewed studies.

3- The scope of reviewed studies is unclear; no mention of search databases, inclusion/exclusion criteria, or study timeline.

4-Despite reviewing numerous proteases, the authors fail to propose specific biomarkers with clinical utility.

5- The review does not explore or discuss conflicting findings.

6- Quantitative information is notably missing for most proteases.

7- The biological significance of the elevated protease levels is not discussed in the context of bladder cancer progression or pathogenesis.

8-There is no discussion of standardization issues across studies (e.g., sample type, assay technique, or patient heterogeneity).

9- The authors fail to correlate protease levels with bladder cancer stages, grades, or prognosis.

10- Key challenges in translating protease biomarkers into clinical diagnostics (e.g., reproducibility, validation, regulatory approval) are ignored.

Reviewer 2 Report

Comments and Suggestions for Authors

This review aims to consolidate existing findings on the role and quantification of proteases in bladder cancer (BC), with a focus on their potential as non-invasive biomarkers. The topic is highly relevant and scientifically valuable, especially in the context of early detection and monitoring of BC, the manuscript requires revision to improve clarity, depth of analysis, and overall impact.

  1. The introduction lacks a structured flow. The clinical significance of bladder cancer should be better contextualized with recent incidence and mortality data (WHO, GLOBOCAN, etc.).
  2. Clearly define the rationale for focusing on proteases. Why are they prioritized over other classes of biomarkers like miRNAs or EVs?
  3. The manuscript currently provides a descriptive rather than critical or comparative assessment of the literature. Consider adding:
  4. A table summarizing studies: type of protease, sample type, method of detection, fold change vs. control, and sample size.
  5. Discussion on inconsistencies across studies, e.g., differences in assay sensitivity, sample preparation, or patient demographics.
  6. Incorporate discussion of emerging technologies for protease detection (e.g., mass spectrometry, biosensors, immunoassays with nanomaterials).
  7. The key finding that most studies lack quantitative data on cathepsins is important. However, this insight should be supported by a systematic breakdown: how many studies included quantification? Which methods were used? What detection limits were reported?
  8. Clarify units (µg/g tissue protein vs. ng/mL in fluids) and provide context—e.g., what threshold levels might be clinically relevant?
  9. Some important recent studies on MMPs and cathepsins in cancer diagnostics seem to be missing. Ensure inclusion of recent (2022–2024) high-quality publications, especially from clinical or translational research journals.
  10. Mention potential for clinical translation or integration into multi-marker diagnostic panels.
  11. The following studies are suggested to evaluate and add in the literature review of manuscript: https://doi.org/10.1016/j.jpba.2023.115937, https://doi.org/10.3389/fonc.2024.1432869,10.1080/17435889.2025.2493037

Reviewer 3 Report

Comments and Suggestions for Authors

The topic addressed by the authors is conceptually interesting, within the field of oncobiology, especially as it highlights an important conclusion: "the reviewed papers do not provide a clear picture concerning protease as bladder cancer biomarkers." In my opinion, this lack of consensus in the literature is itself valuable, as negative or inconclusive findings can guide future research by identifying knowledge gaps and methodological limitations.

In general, the review is supported by the literature. However, the manuscript requires thorough revision in terms of scientific aspects and language quality. The text appears to have been written with limited attention to detail and should be reviewed by a native or fluent English speaker.

Specific comments:

Title: Since the review focuses specifically on cathepsins and matrix metallopoteinases, the paper title is too broad. A more specific title would better inform and attract the appropriate reader.

Introduction: the introduction adequately contextualizes the subject and describes de protease classes that will be discussed. However, regarding incidence, prevalence, and mortality statistics, the authors should follow the most recent GLOBOCAN data. Furthermore, the statement that "Bladder carcinoma (BC) is evaluated as the fifth or tenth most common cancer worldwide" is ambiguous and should be clarified. Any updates made in this section should also be reflected in the "Simple Abstract" and "Abstract."

Additionally, the rational for selecting the two specific classes of proteinases (cathepsins and MMPs) should be explicitly stated and justified.

Cathepsins:

  1. Cathepsin D: (line 88): What does "cancer cytosol" mean? The text is unclear and the authors should explain. Moreover, unlike other cathepsins discussed, no concentration data are provided for this enzyme in tissues, plasma/serum, or urine. The authors should include these data or justify their absence.
  2. b) Cathepsin L (line 118): the mention of “new method” for determining cathepsin L levels should be clarified. What is this method, and how does it improve upon previous approaches?
  3. c) Cathepsin V (lines 127 to 130): there is a repetition in this section that should be eliminated.

Discussion and Tables: the discussion is structured around three tables, which require careful revised, taking into account the bibliographic references, which do not match what is written in the previous text. This is especially critical in tables 2 and 3.

  1. a) Table 2: this table needs significant improvements. The MMP concentration values listed are inconsistent with the narrative descriptions. The concentration values for the various grades of bladder cancer should also appear, and all values should include appropriate statistical descriptors (mean/median +/- standard deviation/standard error). The reference numbers also do not match the data in the description. The authors should correct them.
  2. b) Table 3: in addition to the reference numbers that not matching the description, the lines referring to MMP2 and MMP3 are swapped. The lines referring to CathB should be carefully reviewed, not only because of the concentration units but also because of the values themselves.

Concussions: the conclusions are overly simplistic and would benefit from greater nuance. A more comprehensive synthesis of the findings is needed, including discussion of the limitations of the reviewed studies and potential directions for future research.

Bibliographic references: the reference list should be carefully reviewed according to the journal's guidelines. Some cited works are over 30 years old, so their selection should be justified. Considering the rapid advancement in the field, it is recommended to use more recent references.

Comments on the Quality of English Language

The manuscript requires thorough revision in terms of scientific aspects and language quality. The text appears to have been written with limited attention to detail and should be reviewed by a native or fluent English speaker.